# Impaired cerebrovascular reactivity correlates with reduced retinal vessel density in patients with carotid artery stenosis: Cross-sectional, single center study

Rita Magyar-Stang[1,2]*, Lilla István[3], Hanga Pál[1,2], Borbála Csányi[4], Anna Gaál[4], Zsuzsanna Mihály[5], Zsófia Czinege[5], Péter Sótonyi[5], Horváth Tamás[6], Akos Koller[6,7,8], Dániel Bereczki[1], Illés Kovács[3,9,10], Róbert Debreczeni[1]

1 Department of Neurology, Semmelweis University, Budapest, Hungary, 2 Szentágothai Doctoral School of Neurosciences, Semmelweis University, Budapest, Hungary, 3 Department of Ophthalmology, Semmelweis University, Budapest, Hungary, 4 Faculty of Medicine, Semmelweis University, Budapest, Hungary, 5 Department of Vascular and Endovascular Surgery, Semmelweis University, Budapest, Hungary, 6 Research Center for Sport Physiology, Hungarian University of Sports Science, Budapest, Hungary, 7 Department of Morphology&Physiology, Faculty of Health Sciences, and Translational Medicine Institute, Faculty of Medicine, and ELKH-SE, Cerebrovascular and Neurocognitive Disorders Research Group, Semmelweis University, Budapest, Hungary, 8 Department of Physiology, New York Medical College, Valhalla, NY, United States of America, 9 Department of Ophthalmology, Weill Cornell Medical College, New York, NY, United States of America, 10 Department of Clinical Ophthalmology, Semmelweis University, Budapest, Hungary

* stang.rita@semmelweis.hu

**Data Availability Statement:** All relevant data are within the manuscript and its Supporting Information files.

## Abstract

### Background

The cerebral and retinal circulation systems are developmentally, anatomically, and physiologically interconnected. Thus, we hypothesized that hypoperfusion due to atherosclerotic stenosis of the internal carotid artery (ICA) can result in disturbances of both cerebral and retinal microcirculations. We aimed to characterize parameters indicating cerebrovascular reactivity (CVR) and retinal microvascular density in patients with ICA stenosis, and assess if there is correlation between them.

### Methods

In this cross-sectional study the middle cerebral artery (MCA) blood flow velocity was measured by transcranial Doppler (TCD) and, simultaneously, continuous non-invasive arterial blood pressure measurement was performed on the radial artery by applanation tonometry. CVR was assessed based on the response to the common carotid artery compression (CCC) test. The transient hyperemic response ratio (THRR) and cerebral arterial resistance transient hyperemic response ratio (CAR-THRR) were calculated. Optical coherence tomography angiography (OCTA) was used to determine vessel density (VD) on the papilla whole image for all (VDP-WI$_{all}$) and for small vessels (VDP-WI$_{small}$). The same was done in the peripapillary region: all (VDPP$_{all}$), and small (VDPP$_{small}$) vessels. The VD of superficial

**Funding:** The study was supported by the National Office for Research, Development and Innovation (Project no. NKFIK129277 ("Evaluation of cerebrovascular events in patients with occlusive carotid artery disorders based on morphological and hemodynamic features") has been implemented with the support provided by the Ministry of Innovation and Technology of Hungary from the National Research, Development, and Innovation Fund. AK: NKFI-1 K OTKA 132596 K-19, TKP2021-EGA-37 of MIT of Hungary-NRDI TKP2021-EGA funding and HAS/MTA Post-Covid 2021-34. DB: Ministry of Innovation and Technology of Hungary from the National Research, Development and Innovation Fund (Grant No: TKP2021-EGA/TKP2021-NVA/TKP2021-NKTA). The funders had no role in study design, data collection and analysis, decision to publish, or preparation of the manuscript.

**Competing interests:** The authors have declared that no competing interests exist.

$(VDM_{spf})$ and deep $(VDM_{deep})$ macula was also determined. Significance was accepted when $p < 0.05$.

## Results

Twenty-four ICA stenotic patients were evaluated. Both CVR and retinal VD were characterized. There was a significant, negative correlation between CAR-THRR (median = -0.40) and $VDPP_{small}$ vessels (median = 52%), as well as between $VDPP_{all}$ vessels (median = 58%), and similar correlation between CAR-THRR and $VDP-WI_{small}$ (median = 49.5%) and between $VDP-WI_{all}$ (median = 55%).

## Conclusion

The significant correlation between impaired cerebrovascular reactivity and retinal vessel density in patients with ICA stenosis suggests a common mechanism of action. We propose that the combined use of these diagnostic tools (TCD and OCTA) helps to better identify patients with increased ischemic or other cerebrovascular risks.

## Introduction

Most of the total deaths in the countries of the world, regardless of their economic development, are due to cardiovascular diseases, and their most common pathological basis is atherosclerosis [1]. This progressive, generalized vascular disease is probably inevitable since aging is the number one and uncontrollable risk factor for its development [2–6]. In the asymptomatic population, atherosclerosis can be detected in 50% over the age of 40, and the rate of consequent carotid stenosis is 10% over the age of 60 [7]. The considerable portion of cerebral ischemic events (~30%) results from atherosclerotic steno-occlusive disease of the internal carotid artery (ICA) [1, 8, 9], which can be prevented and treated if patients at high ischemic risk are identified in time. The most serious complication of cerebral circulation is due to hypoperfusion resulting in cerebral and retinal ischemia [10–12], which could lead to disability, cognitive and retinal dysfunction as shown by clinical [6, 13–15] and experimental studies [16–21]. These parallel pathological events are logical consequences of the fact that the cerebral and retinal circulations are developmentally, anatomically, and physiologically interconnected [8, 22, 23]. However, there is few if any study investigating the cerebral and retinal vascular alterations simultaneously in patients with atherosclerotic ICA stenosis, which would however show the general presence of atherosclerotic vascular disease and facilitate the correct diagnosis and the consequent treatments. On the basis of the above we hypothesized that in patients with atherosclerotic ICA stenosis–which is likely an indicator of systemic atherosclerotic vascular diseases–both cerebral and retinal circulations are affected [15, 24] and there is a correlation between the functional responses of cerebral vessels. Fig 1 illustrates the common origin of the two vascular networks.

### Aim

The aim of the present study was to characterize the parameters indicating cerebrovascular reactivity and retinal vascular density changes and assess if correlation exist between them in older adults with atherosclerotic carotid stenosis.

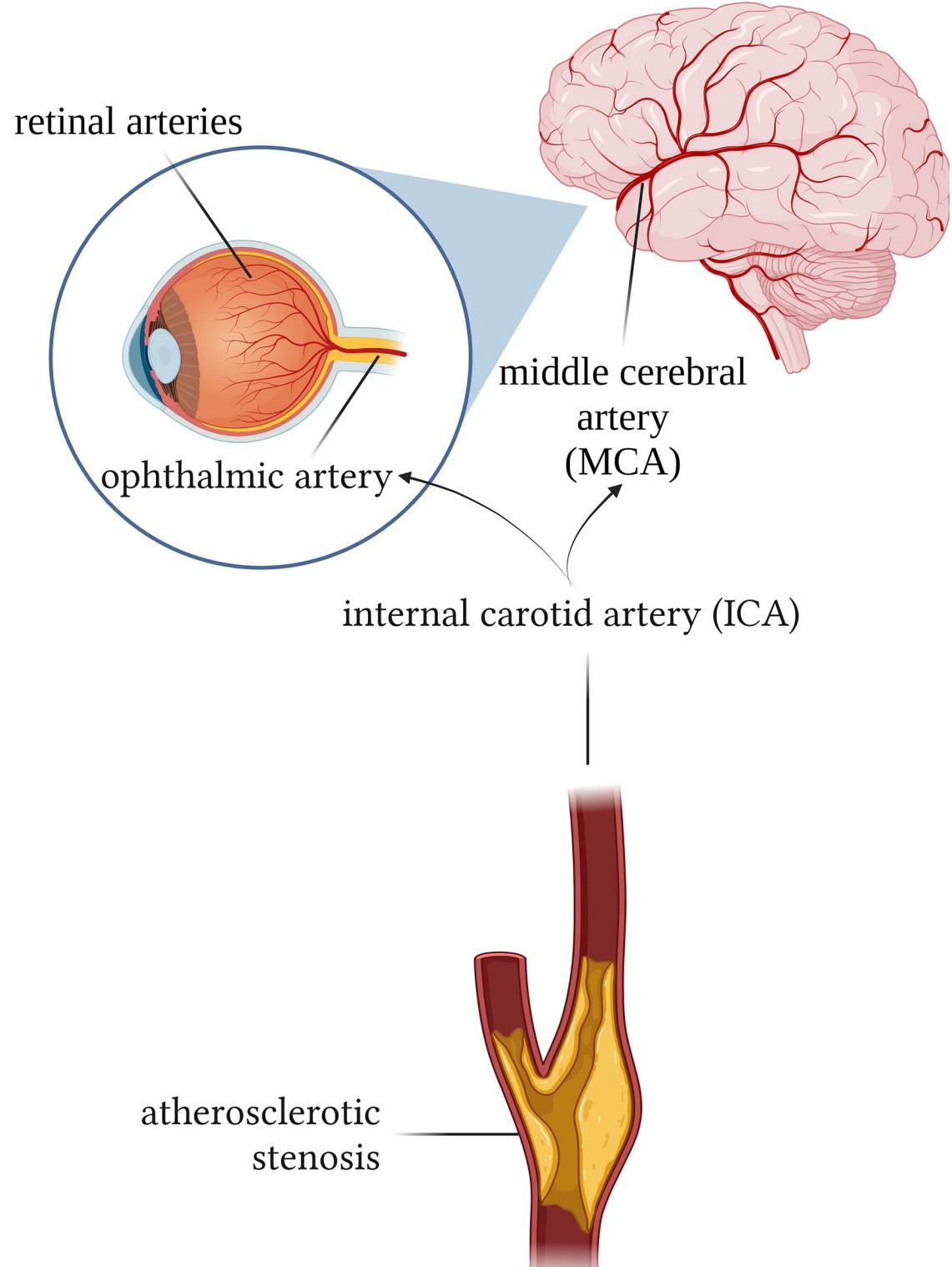

**Fig 1. Schematic depiction of cerebral and retinal circulation affected of atherosclerotic stenosis of the internal carotid artery.**
ICA = internal carotid artery; MCA = middle cerebral artery (created with www.biorender.com).

## Materials and methods

This cross-sectional study was approved by the Regional, Institutional Scientific Research Ethics Committee of Semmelweis University (SE RKEB permit number: 256/2018) and registered on the ClinicalTrials.gov website (Reg No: NCT03840265). The study was conducted according to the guidelines of the Declaration of Helsinki. Patients with significant ICA stenosis were evaluated at the Department of Vascular and Endovascular Surgery of Semmelweis University after providing them with detailed oral information and gaining their written consent, were prospectively and consecutively enrolled between 01.01.2019 and 30.09.2021. and data processing began after patient enrollment. Authors participating in data curation, project administration and investigation had access to information that could identify individual participants during or after data collection. Patients underwent CT angiography as part of standard-of-care diagnostic evaluation using a routine clinical imaging protocol. Patients with significant (≥70%), symptomatic (with ipsilateral, hemispheric, non-lacunar infarct or transient ischemic attack in the past 6 months) and asymptomatic (without ipsilateral, hemispherical infarct) ICA stenosis were screened for enrolment [25].

Inclusion criteria for the study were threefold: significant carotid artery stenosis (≥ 70%) was determined by 256-slice computed tomography angiography (CTA) (Brilliance iCT 256, Philips Healthcare, Best, Netherlands) based on the North American Symptomatic Carotid Endarterectomy Trial (NASCET) criteria [25] and planned endarterectomy. Exclusion criteria for the TCD examination were atrial fibrillation, extensive embolic plaques in the common carotid artery prone to fragment when compressed, and carotid sinus hyperesthesia. Patients with macular degeneration, glaucoma, vitreomacular diseases, previous intraocular anti-VEGF (vascular endothelial growth factor) injection, clinically significant media opacity, and nystagmus were excluded from OCTA examination. Demographic and medical history data of the patients were prospectively collected.

During the study period, 89 of 108 patients with significant ICA stenosis agreed to participate in the study. Among these, TCD examination could not be performed on 18 patients due to a completely missing insonation window, and on further 15 patients TCD evaluation was not possible due to a non-optimal signal-to-noise ratio. Altogether 32 patients were excluded from the study group due to the OCTA exclusion criteria. Finally, we analyzed the TCD and OCTA results of 24 patients with significant ICA stenosis.

### Transcranial Doppler (TCD) study protocol

TCD was performed as previously described [26–28] utilizing a novel data processing method (S1 Appendix). During TCD measurements, continuous, non-invasive, beat-to-beat arterial blood pressure (ABP) monitoring was performed with radial artery applanation tonometry (Colin-BP508, Hayashi Komaki Aichi, Japan). Calibration to mmHg before and after every common carotid artery compression test was made by the same equipment's sphygmomanometer.

**Data processing.**    A detailed description of the data processing can be found in the supplementary information (S2 Appendix).

### Common carotid artery compression test–CCC test [29–32]

To assess the CVR we have used CCC test, which consists of a temporal (10 s) manual compression of the common carotid artery, while the BFV is continuously measured and registered in the ipsilateral MCA with TCD. During CCC, a significant decrease in BFV occurs in the ipsilateral MCA, and when compression is ended, a rapid, transient increase in BFV can be measured: this phenomenon is the transient hyperemic response (THR) [31, 32].

Fig 2 shows an original registration of the changes in blood flow velocity measured by TCD, before, during and after CCC (Fig 1).

The CCC test was performed in a standardized manner for 10 seconds [31] (S3 Appendix).

**Parameters of blood flow velocity changes in response to CCC.** Transient hyperemic response ratio (THRR): expresses the change in systolic BFV measured directly after the release of CCC compared to the baseline value.

$$\textbf{THRR} = \frac{V_{S2} - V_{S1}}{V_{S1}}$$

$V_{S1}$ = average systolic BFV of the baseline; $V_{S2}$ = systolic BFV maximum after the release of the compression (Fig 2).

Cerebral arterial resistance—transient hyperemic response ratio (CAR-THRR): cerebral arterial resistance (CAR) refers to the reciprocal value of the change in mean $BFV_{mean}$ in MCA ($BFV_{mean\ MCA}$) per unit change in $ABP_{mean}$.

$$\textbf{CAR} = \frac{ABP_{mean}}{BFV_{mean\ MCA}}$$

CAR cannot be calculated during compression (dashed line on Fig 2/B), but at baseline and after CCC release. Similar to THRR, CAR-THRR expresses the change in CAR measured at release of CCC compared to the baseline CAR value.

$$\textbf{CAR} - \textbf{THRR} = \frac{CAR_2 - CAR_1}{CAR_1}$$

Where $CAR_1$ = average of baseline CAR; $CAR_2$ = CAR at the release of the CCC.

CAR-THRR therefore also takes into account the simultaneous change in arterial blood pressure. If cerebral resistance vessels dilate because of reduced MCA perfusion due to carotid compression, then resistance to flow decreases, hence the value is negative. A less negative value corresponds to a greater decrease in resistance during compression, indicating a greater cerebrovascular reactivity. A lower value for THRR and a higher value for CAR-THRR indicate decreased cerebrovascular reactivity (CVR) (Fig 2).

## Optical coherence tomography angiography (OCTA) study protocol

OCTA is a fast, non-invasive technique, that allows visualization of blood flow in different layers of the retina and choroid without the need of any contrast agent injection. OCTA measures the movement of red blood cells in the blood vessels, which provides information about the perfusion of the different tissues [33–36]. In the present study, on the same patients undergoing TCD measurements OCTA examination was performed with the AngioVue OCTA system as previously described [22], using the SSADA (split-spectrum amplitude-decorrelation angiography) software algorithm (RTVue XR Avanti with AngioVue, Optovue Inc, Fremont, CA, USA) (Fig 3). The software evaluated the image quality (scan quality—SQ) of the OCTA recordings on a 10-point scale, and only test results with an SQ value exceeding 5 points were included in the study [37]. Images containing motion artifacts (such as white line artifacts, vessel discontinuities, vessel doubling, or noise), segmentation errors, or projection artifacts were excluded [37]. Three OCTA examinations of the macular, papillary, and peripapillary areas were performed on each patient, and the best image quality was analyzed during data processing.

**OCTA variables.** One of the parameters that characterize the retinal vascular network is the vessel density (VD) [33, 38–40]. This value expresses the vascularity of the area examined

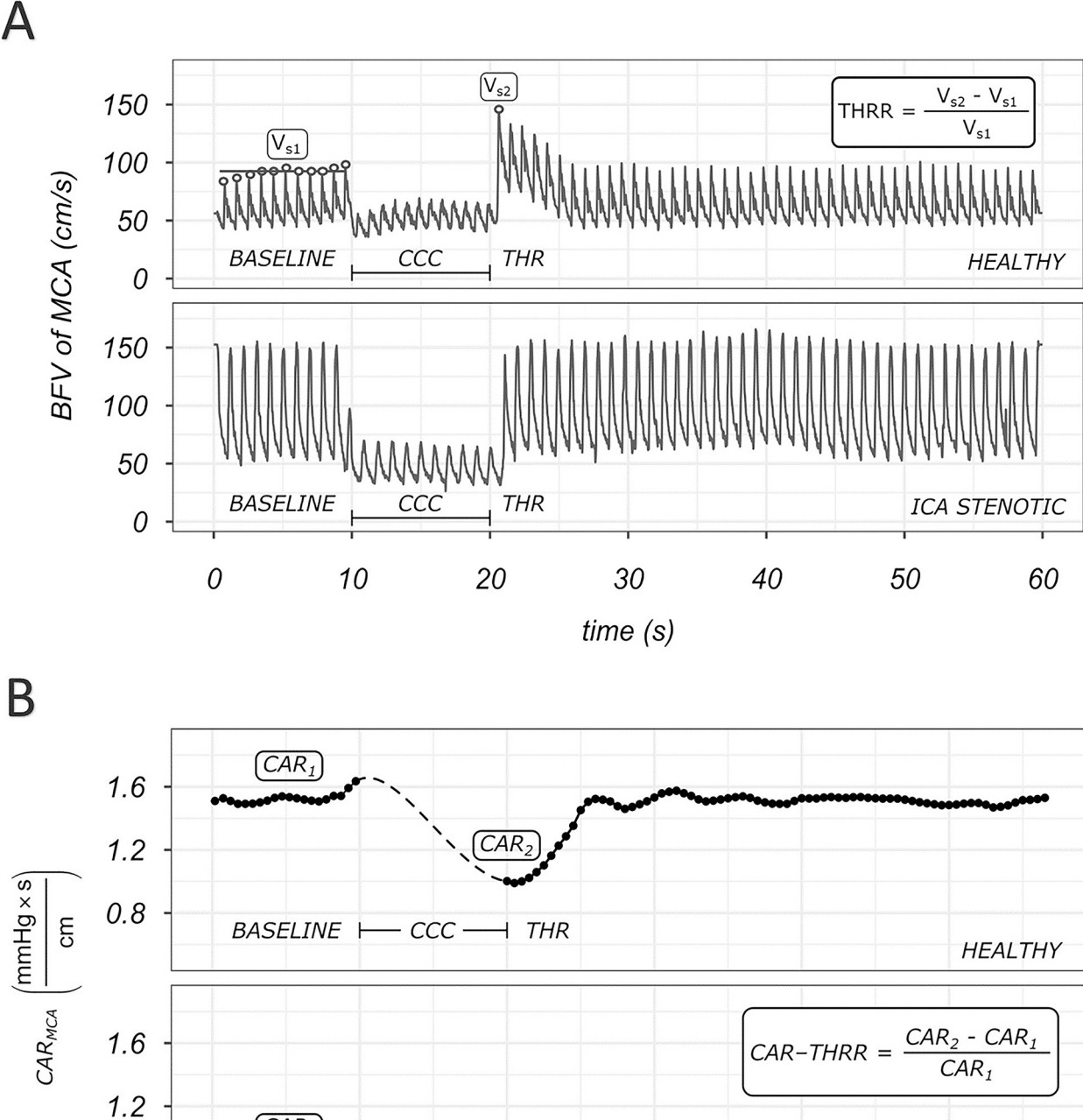

**Fig 2. Original records of the common carotid artery compression test. A (upper panel):** Original figure shows changes in blood flow velocity measured by TCD of the ipsilateral MCA, before, during, and after common carotid artery compression. The increase of BFV above baseline after release of compression is termed transient hyperemic response (THR). The record shows an increase in BFV after the release of compression above baseline, indicating the THR of a healthy volunteer. In the inset: the calculation of the THRR is depicted. BFV of MCA = blood flow velocity of the ipsilateral MCA; CCC = common carotid artery compression; Vs1 = average of systolic BFV of the baseline; Vs2 = maximum systolic BFV after the

release of the compression. **A (lower panel):** Original record shows changes in BFV measured by TCD in the ipsilateral MCA before, during and after CCC. The lack of THR in patients with severe ICA stenosis indicate impaired cerebrovascular reactivity. **B (upper panel):** Calculated cerebral arterial resistance (CAR) of the ipsilateral MCA and its changes before and after the common carotid artery compression. The record shows a decrease in CAR after the release of compression, indicating the transient hyperemic response (THR) of a healthy volunteer. **B (lower panel):** minimally changed CAR and the lack of THR in patient with severe ICA stenosis indicating impaired cerebrovascular reactivity. Since ABP can be considered as arterial perfusion pressure before and after but not during CCC, the dashed line indicates the hypothetical change in CAR. Calculation of CAR is depicted in the inset. CAR-THRR = cerebral arterial resistance–transient hyperemic response ratio; $CAR_1$ = average of baseline CAR; $CAR_2$ = CAR at the release of the compression.

in the skeletonized image. The VD was determined on the papillary area (P) by the software defined 4.5×4.5 mm whole image (whole image—WI) for all vessels ($VDP\text{-}WI_{all}$) and selectively for only the small vessels ($VDP\text{-}WI_{small}$) (S4 Appendix). The peripapillary region (PP) was defined by the software as an area enclosed by a 2.5 mm inner ring and a 4.5 mm outer ring, centered on the optic disc. Similarly, on the PP, VD was also determined for all vessels ($VDPP_{all}$) and selectively for small vessels ($VDPP_{small}$). Another region examined was the superficial ($VDM_{spf}$) and deep ($VDM_{deep}$) vascular network of the macula in a 3×3 mm area.

## Statistical analysis and data processing

Due to the non-normal distribution of the continuous variables and the relatively small sample size, non-parametric tests were performed: Spearman's correlation test and the Mann-Whitney U test were used. Missing data were excluded from the calculations. Statistical analysis and data visualization were performed with the Prism Graph Pad software (Version 8.0.1., San Diego, CA USA). P<0.05 was considered as statistically significant. The normality test was performed with the Kolmogorov-Smirnov test, based on which the data showed a non-normal distribution. The minimum sample size was determined by statistical power calculation (power 0.80; p = 0.05) using data from previous studies at our institution and the method proposed by Hulley et al. for correlation analyses [41]. The minimum number of eyes to be

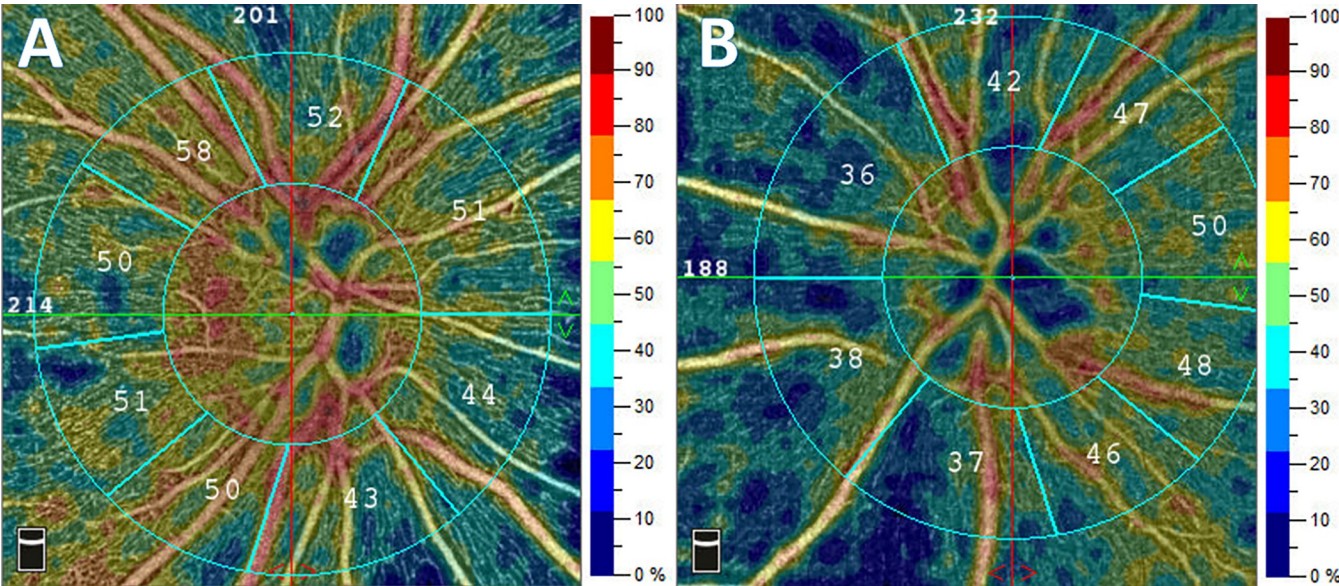

**Fig 3. Color coded OCTA images of vascular networks of the papilla.** Comparison of images of a young, healthy volunteer (A) and an older patient with significant internal carotid artery (ICA) stenosis (B) shows substantial differences in the vascular density (VD) of their papilla. The color scale on the right indicates the percentage of vessel density.

included in this study was calculated to be 22 eyes to provide sufficient power for bivariate correlation analyses.

## Results

Demographic and medical history data of the patients are summarized in Table 1. In Table 2 the TCD and OCTA data characteristics are summarized.

### Cerebrovascular reactivity and calculated cerebral arterial resistance

In order to illustrate changes in cerebrovascular reactivity a CCC test-induced transient hyperemic response of a healthy volunteer with preserved cerebrovascular reactivity and a patient with severe ICA stenosis and impaired cerebrovascular reactivity are shown in Fig 2.

In order to illustrate changes in arterial resistance a transient hyperemic response of a healthy volunteer with preserved resistance change and a patient with severe ICA stenosis with decreased cerebral arterial resistance change are shown in Fig 2.

To contrast these results to healthy vascular system in the present study five age-matched control participants (mean age = 63.2 ± 7.1) with an absence of atherosclerotic carotid artery disease and other major vascular risk factors were recruited. Statistical analysis showed a significant difference in CAR-THRR between the control and study group (Mann-Whitney U test, p = 0.045).

### Vessel density

To illustrate the correspondence between functional cerebrovascular changes and changes in the blood flow in retinal vasculature in Fig 3, retinal microvascular images of a healthy volunteer (left) and that of the same patient included in Fig 2 with severe ICA stenosis (right) are depicted. It can be well appreciated that, compared to healthy subject, both total and macular vessel densities are substantially reduced in the ICA stenotic patient (Fig 3).

### Correlations between TCD and OCTA variables

Among the variables of the TCD and OCTA, there was a significant, negative correlation between the cerebral arterial resistance—transient hyperemic response ratio (CAR-THRR) and peripapillary vessel density of small vessels ($VDPP_{small}$) (p = 0.003; Spearman r = -0.57) (Fig 4/A), and of all vessels ($VDPP_{all}$) (p = 0.01; Spearman r = -0.48) (Fig 4/B). We also found a significant, negative correlation between the cerebral arterial resistance—transient hyperemic response ratio (CAR-THRR) and vessel density of small vessels in the papilla whole image ($VDP-WI_{small}$) (Spearman correlation test, p = 0.01; Spearman r = -0.52) (Fig 4/C) and

**Table 1. Baseline characteristics of the study group.**

| Demographic data | Value |
|---|:---:|
| Age (mean±SD) years | 69.4 ± 6.8 |
| Sex (male/female) | 18/6 |
| Hypertension (n/%) | 24/100 |
| Diabetes mellitus (n/%) | 7/29 |
| Smoking (n/%) | 8/33 |
| Contralateral ICA steno-occlusive disease (n/%) | 7/29 |
| Symptomatic ICA stenosis (n / %) | 6/25 |

ICA = internal carotid artery

**Table 2. Baseline characteristics of TCD and OCTA data in older patients with ICA stenosis.**

| TCD data | mean ± SD (n = 24) |
|---|---|
| baseline mean ABP (mmHg) | 103.8 ± (18.0) |
| baseline systolic BFV (cm/s) | 82.3 ± (20.1) |
| baseline mean BFV (cm/s) | 54.1 ± (15.0) |
| baseline CAR (mm Hg * s * cm$^{-1}$) | 2.1 ± (0.8) |
| THRR (-) | 0.2 ± (0.2) |
| CAR-THRR (-) | -0.2 ± (0.1) |
| **OCTA data (%)** | **mean ± SD (n = 24)** |
| VDPP$_{small}$ | 51.2 ± (2.7) |
| VDPP$_{all}$ | 57.7 ± (2.6) |
| VDP-WI$_{small}$ | 48.9 ± (2.9) |
| VDP-WI$_{all}$ | 55.2 ± (2.3) |
| VDM$_{spf}$ | 43.6 ± (4.4) |
| VDM$_{deep}$ | 48.1 ± (3.9) |

TCD = transcranial Doppler sonography; ABP = arterial blood pressure; BFV = blood flow velocity; CAR = cerebral arterial resistance; THRR = transient hyperemic response ratio; CAR-THRR = cerebral arterial resistance transient—hyperemic response ratio; OCTA = optical coherence tomography angiography; M = macular region; P = papilla; PP = peripapillary region; WI = whole image; spf = superficial; VD = vessel density; WI = whole image; SD = standard deviation

of all vessels (VDP-WI$_{all}$) (Spearman correlation test, p = 0 .02; Spearman r = -0.45). (Fig 4/D) (Fig 4). There were no correlations among the other OCT and TCD variables (Table 3).

## Discussion

The salient findings of the present study are that in patients with significant ICA stenosis there were 1) functional deficit of cerebrovascular responsiveness, tested by common carotid artery compression test, 2) severely reduced retinal vessel density and 3) strong correlations between the cerebrovascular functional and retinal vessel density variables. These findings suggest that common alterations and pathomechanisms underlie the atherosclerotic changes in the cerebral and retinal vasculature identified for the first time in ICA stenotic patients by simultaneous use of functional TCD and OCTA modalities.

### Hemodynamic basis of reduced cerebrovascular reactivity

To test our hypothesis, we used the non-invasive transcranial Doppler technique (TCD) known to have excellent temporal sensitivity [42–45] to follow the changes in blood flow velocity in the middle cerebral artery (MCA) in various conditions [43, 46]. Assuming that neither the position of the TCD transducer, nor the diameter of the vessel examined changes during a cerebral vasoactive stimulus, the change in flow velocity that occurs is proportional to the change in the cerebral blood flow [47]. Previous studies have shown that the detailed analysis of envelope fitted to the maximum of the flow time-frequency (velocity) power spectrum after fast-Fourier transformation (shape, flow direction, average velocity) created from the Doppler frequency shift value is suitable for the analysis and quantification of hemodynamic changes, such as cerebrovascular reactivity (CVR) [31, 44, 48]. The response to CCC stimulus can be divided into two phases with changes in opposite directions. The vasodilation induced by manually performed common carotid artery compression evokes considerable hypoperfusion, resulting in an increase in MCA blood flow velocity after the cessation of carotid compression,

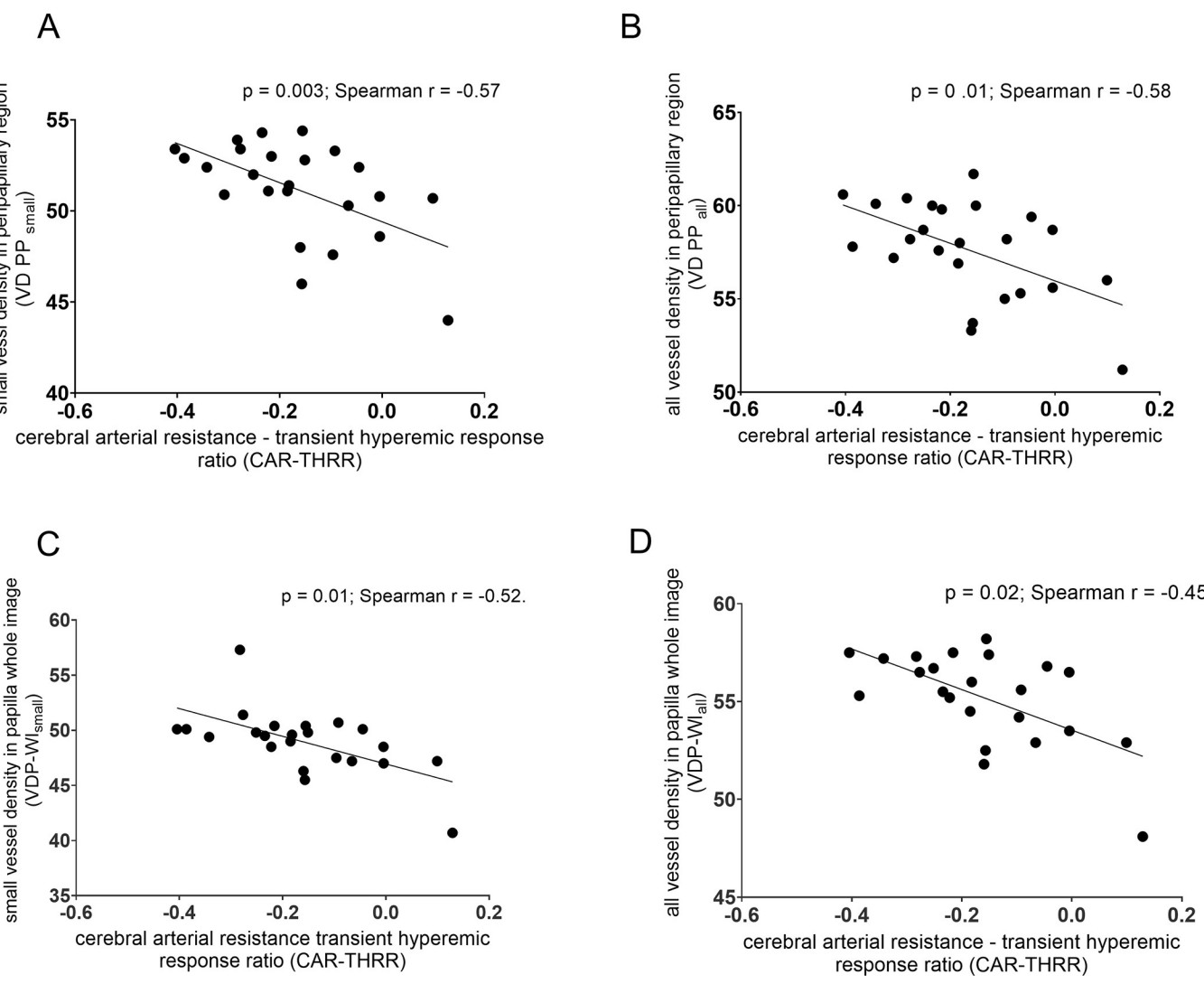

**Fig 4. Significant negative correlations between the calculated cerebral arterial resistance—transient hyperemic response ratio and vessel density of the small and all vessels in the peripapillary/ papilla whole image region in older patients with ICA stenosis.** CAR-THRR = cerebral arterial resistance—transient hyperemic response ratio; VD = vessel density; PP = peripapillary region; P-WI = papilla whole image.

which exceeds the baseline value under intact control. Cerebral arterioles compensate to their maximum vasodilation capacity with maintained pressure autoregulation, and tissue perfusion remains sufficient. This reactive hyperperfusion then results in an opposite process, hyperperfusion-induced vasoconstriction, when MCA cerebral blood flow velocity values return to the baseline, this phenomenon is probably not passive [49, 50]. When there was an immediate and significant increase in MCA blood flow velocity after cessation of carotid compression, i.e., the THRR was high, vasoconstriction induced by increased flow also occurred rapidly. The absence or delay of transient hyperemia indicates a decrease in reactivity. In previous studies, the index calculated from THRR BFV values proved to be suitable for characterizing reduced cerebral vasoreactivity [29–32]. However, in our correlation analysis, no close relationship was seen with the mentioned BFV index and the OCT values. When another index that could determine with the CCC test was used, we found a significant correlation. This index was the CAR-THRR. In the complex evaluation of cerebrovascular reactivity tests, many previous

**Table 3. Spearman correlation tests of OCTA and TCD variables in older patients with ICA stenosis.**

| TCD variables | OCTA variables | Spearman R | p value |
|---|---|---|---|
| **CAR-THRR** | $VDPP_{small}$ | -0.57 | **0.003** |
| THRR | | 0.18 | 0.39 |
| **CAR-THRR** | $VDPP_{all}$ | -0.48 | **0.01** |
| THRR | | 0.19 | 0.38 |
| **CAR-THRR** | $VDP-WI_{small}$ | -0.52 | **0.01** |
| THRR | | 0.18 | 0.37 |
| **CAR-THRR** | $VDP-WI_{all}$ | -0.45 | **0.02** |
| THRR | | 0.20 | 0.34 |
| CAR-THRR | $VDM_{spf}$ | 0.08 | 0.68 |
| THRR | | -0.18 | 0.40 |
| CAR-THRR | $VDM_{deep}$ | 0.18 | 0.39 |
| THRR | | -0.14 | 0.50 |

CAR-THRR = cerebral arterial resistance transient hyperemic response ratio; M = macular region; P = papilla; PP = peripapillary region; spf = superficial;

THRR = transient hyperemic response ratio; VD = vessel density; WI = whole image

studies emphasize the consideration of arterial blood pressure [49–52]. While the THRR is based only on the BFV change measured in the MCA, the CAR-THRR variable takes into account the simultaneous arterial blood pressure changes in addition to the BFV changes. Accordingly, the change in cerebral arterial resistance of the MCA after the cessation of manually induced (CCC) hypoperfusion, which was used to characterize the cerebral vasoreactivity, can be determined more precisely. Therefore, the CCC test can be considered a dynamic autoregulation testing method, as changes in ABP and BVF are significant within a short time. The resistance calculation of the cerebral vessels is advantageous because it expresses not only blood flow velocity but also simultaneous arterial blood pressure changes. Henceforward, in purpose of CVR characterization, parallel ABP and HR registration is considered essential because these factors are not insignificant when evaluating cerebral vasoreactivity. Normally, the CAR value at the end of CCC test is low due to maximal vasodilation, so the CAR-THRR value is negative. Thus, the decrease in blood flow velocity resistance developed after the cessation of the CCC, which was used to quantify the vasoreactivity of the brain tissue perfused by the MCA, can more accurately characterize the cerebral vascular reserve.

## Role of regional differences in retinal vasculature

In the present study a correlation was only found between impaired cerebrovascular reactivity and hypoperfusion resulting from decreased circulation of papillary and peripapillary regions, but not of the macula region. The difference can be explained by the fact that the anatomical blood supply and regulatory mechanisms of the various intraocular regions are different [53, 54]. The arterial blood supply of the eye is provided by the ICA via several branches of the ophthalmic artery (central retinal artery—ensuring blood supply of the superficial layers of the retina, the short and long posterior ciliary arteries, and the anterior ciliary arteries—mainly suppling the outer retinal layers and eye structures and forming the choroid) [53, 54]. In the circulation of the macula, the branches of the central retinal artery are decisive, the macula normally has an avascular zone at the fovea where these branches are not present [55]. Due to the high metabolic demand of the macula, regulation of macular blood flow is controlled primarily by local metabolic mechanisms affecting the central retinal artery branches [56, 57]. In contrast, in the peripapillary circulation the choroid plays an important role. The short

posterior ciliary arteries branch into terminal arterioles to form the arterioles of the dense outer layer of the choroid [58]. Previous studies suggest local metabolic local control does not seem to play a role in choroidal blood flow regulation [59, 60]. The choroid is richly innervated, and thus neural control mechanisms are the most relevant regulating factors [61]. The papilla is supplied by the peripapillary choroid composed of the scleral short posterior ciliary system and the recurrent choroidal arteries. Regulation, similarly to the peripapillary region, is controlled by neuronal mechanisms [62, 63].

Many of the findings of the present study are consistent with previously reported results in the literature. Similarly to our study, Zhang et al. analyzed the correlation of OCT and TCD procedures in cerebrovascular patients with ischemic and hemorrhagic stroke. They found a significant, positive correlation between baseline MCA blood flow velocity values and retinal vessel density, however, functional TCD test was not used in their study [64]. Bettermann et al. investigated retinal vasoreactivity in patients with ischemic white matter disease using a high-frequency flashing light stimulation method and showed that retinal structural microvascular damage is associated with functional impairment [65], the extent of which correlates with cerebrovascular reactivity based on the TCD-registered breath-holding test, i.e. decreased blood flow [66] velocity response in the MCA. The great advantage of that study was that it combined two functional methods, which are not performed routinely in patients [65, 67]. In line with our findings, a correlation between MCA baseline blood flow velocity parameters and retinal vessel caliber variability was found in patients with intracranial stenosis using a semi-automated computer assisted program (Singapore I Vessel Assessment) [68].

Optical coherence tomography angiography (OCTA) is known to be suitable for providing images of the retinal vascular network with high accuracy [69–71]. The OCTA method is based on the motion-contrast phenomenon, i.e., the position of the moving structures—the red blood cells—constantly changes in the layers of the retinal vascular network, while no such change occurs in poorly perfused or not at all perfused areas [72]. OCTA is a fast and easily reproducible technique, providing qualitative and quantitative results on various regions of the retinal microcirculation based on the movements of red blood cells [36]. The great advantage of the OCTA examination is that it was previously found to be suitable for monitoring retinal circulation changes after ICA revascularization, as several studies reported a significant improvement in retinal circulation after ICA revascularization not only on the ipsilateral but also on the contralateral side [22, 73–76]. Regarding the OCTA studies previous studies on the same patient group described the decrease of retinal vessel density in this patient group [22].

## Strengths of the study

The main strength of our study lies in the fact that this is the first study that analyzes the correlation of functional TCD and OCTA modalities in ICA stenotic patients. The strength of the methodology of this study is its multimodal nature, which, in addition to changes in blood flow velocity, also continuously detects simultaneous changes in blood pressure, and from their rapidly and significantly changing ratio, enables accurate hemodynamic characterization. Another advantage is that our study compares two relatively easily performed, non-invasive modalities (TCD and OCTA), both of which are available in clinical routine. In the international guidelines, the percentage of the ICA stenosis and the symptoms are the determining factors in the treatment strategy of patients [66] although, no uniform consensus has developed to identify additional factors contributing to the risk of cerebral ischemia. For this reason, an additional advantage of the present study is that high-risk ICA stenotic patient can be identified by the combined use of the two modalities.

## Limitations

The limitation of our study is the low number of patients, primarily due to the missing or very narrow temporal acoustic insonation window in 20–30% of the study group, likely due to the age of and gender of the patients. In our study, the rate of occurrence of an inadequate temporal insonation window is higher than in literature data, which we explain by the relatively high average age of the patients and our strict TCD examination protocol. It is of note that both TCD and OCTA examinations have technical limitations [33, 42, 44, 69, 70, 77]. However, it is also considered as the strength of our study, that we applied two very specific non-invasive techniques in the same patients. These, however, came with some limitations, but on the other hand—we believe—it makes our study unique. However, loss of patients was mainly due to strict inclusion criteria of the study protocol. Also, 17% of patients decline the examinations, because it would require extended hospital stay. In addition, the study period coincided with the COVID-19 pandemic affecting Hungary and coinciding with lockdown periods. These factors further reduced the number of potentially eligible patients in the present study.

## Conclusion

The present study showed, for the first time, significant correlations between impaired cerebrovascular reactivity and reduced retinal vessel density in patients with internal carotid artery stenosis, suggesting that similar atherosclerotic changes occur, with similar mechanisms of actions in the two vascular systems, most likely due to the chronic hypoperfusion of the brain. The translational/clinical importance of these findings is that the combined use of these diagnostic tools (TCD and OCTA) may help to better identify patients with increased low-flow ischemic risk due to atherosclerosis and/or initiate appropriate treatments. Despite the mentioned limitations, based on the proven significant correlation, the two test methods complement and may even replace each other, so when one method cannot be used for technical reasons, the other can provide clinical data of similar value. Nevertheless, further studies with a larger number of patients are necessary to confirm the conclusion of present findings.

## Supporting information

**S1 Checklist. STROBE statement—checklist of items that should be included in reports of observational studies.**
(DOCX)

**S1 Appendix. Transcranial Doppler (TCD) study protocol.** The text summarizes the technical description of TCD measures.
(DOCX)

**S2 Appendix. Data processing.** The text summarizes the software data processing.
(DOCX)

**S3 Appendix. Common carotid artery compression test.** The text summarizes the description of the standardized common carotid artery compression test.
(DOCX)

**S4 Appendix. OCTA small vessel criteria.** The text summarizes the description of the software-based small vessel definition.
(DOCX)

**S1 File. Dataset of the study.**
(PDF)

**S2 File. English ethical approval.**
(PDF)

## Author Contributions

**Conceptualization:** Rita Magyar-Stang, Zsuzsanna Mihály, Péter Sótonyi, Horváth Tamás, Róbert Debreczeni.

**Data curation:** Rita Magyar-Stang, Lilla István, Hanga Pál, Borbála Csányi, Zsuzsanna Mihály, Zsófia Czinege, Horváth Tamás.

**Formal analysis:** Rita Magyar-Stang, Lilla István, Borbála Csányi, Anna Gaál, Zsuzsanna Mihály, Zsófia Czinege, Péter Sótonyi, Horváth Tamás, Akos Koller, Róbert Debreczeni.

**Funding acquisition:** Péter Sótonyi.

**Investigation:** Rita Magyar-Stang, Lilla István, Hanga Pál, Borbála Csányi, Anna Gaál, Zsuzsanna Mihály, Horváth Tamás.

**Methodology:** Rita Magyar-Stang, Zsuzsanna Mihály, Horváth Tamás, Róbert Debreczeni.

**Project administration:** Rita Magyar-Stang, Hanga Pál, Zsuzsanna Mihály, Horváth Tamás.

**Resources:** Péter Sótonyi, Akos Koller, Dániel Bereczki, Róbert Debreczeni.

**Software:** Rita Magyar-Stang, Horváth Tamás.

**Supervision:** Péter Sótonyi, Akos Koller, Dániel Bereczki, Illés Kovács, Róbert Debreczeni.

**Validation:** Zsuzsanna Mihály, Dániel Bereczki, Róbert Debreczeni.

**Visualization:** Rita Magyar-Stang, Horváth Tamás.

**Writing – original draft:** Rita Magyar-Stang.

**Writing – review & editing:** Lilla István, Zsuzsanna Mihály, Péter Sótonyi, Horváth Tamás, Akos Koller, Dániel Bereczki, Illés Kovács, Róbert Debreczeni.

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
