## [Decision Letter · Decision Letter 0]

4 Jun 2023

PONE-D-23-12728Impaired cerebrovascular reactivity correlates with reduced retinal vessel density in patients with carotid artery stenosisPLOS ONE

Dear Dr. Magyar-Stang,

Thank you for submitting your manuscript to PLOS ONE. After careful consideration, we feel that it has merit but does not fully meet PLOS ONE’s publication criteria as it currently stands. Therefore, we invite you to submit a revised version of the manuscript that addresses the points raised during the review process.

We look forward to receiving your revised manuscript.

Kind regards,

Ayman Elnahry

Academic Editor

PLOS ONE

“Ministry of Innovation and Technology of Hungary from the National Research, Development and Innovation Fund (Grant No: TKP2021-EGA/TKP2021-NVA/TKP2021-NKTA). “

“PS: The study was supported by the National Office for Research, Development and Innovation (Project no. NKFI-K129277 ("Evaluation of cerebrovascular events in patients with occlusive carotid artery disorders based on morphological and hemodynamic features") has been implemented with the support provided by the Ministry of Innovation and Technology of Hungary from the National Research, Development, and Innovation Fund.

ÁK: NKFI-1 K OTKA 132596 K¬_19, TKP2021-EGA-37 of MIT of Hungary-NRDI TKP2021-EGA funding and HAS/MTA Post-Covid 2021-34. The funders had no role in study design, data collection and analysis, decision to publish, or preparation of the manuscript.”

3. We note that Figure 2 in your submission contain copyrighted images. All PLOS content is published under the Creative Commons Attribution License (CC BY 4.0), which means that the manuscript, images, and Supporting Information files will be freely available online, and any third party is permitted to access, download, copy, distribute, and use these materials in any way, even commercially, with proper attribution. For more information, see our copyright guidelines: http://journals.plos.org/plosone/s/licenses-and-copyright.

Reviewers' comments:

Reviewer's Responses to Questions

**Comments to the Author**

1. Is the manuscript technically sound, and do the data support the conclusions?

Reviewer #1: Yes

Reviewer #2: Partly

Reviewer #3: Yes

Reviewer #4: Partly

2. Has the statistical analysis been performed appropriately and rigorously? 

Reviewer #1: Yes

Reviewer #2: Yes

Reviewer #3: Yes

Reviewer #4: No

3. Have the authors made all data underlying the findings in their manuscript fully available?

Reviewer #1: Yes

Reviewer #2: No

Reviewer #3: Yes

Reviewer #4: Yes

4. Is the manuscript presented in an intelligible fashion and written in standard English?

Reviewer #1: Yes

Reviewer #2: Yes

Reviewer #3: Yes

Reviewer #4: No

5. Review Comments to the Author

Reviewer #1: the present is an interesting study aiming to correlate with high tecnhology the relationship between cerebral and retinal circulation

Some issues need to be addressed

title: authors should add if the present is a single center study or not

Abstract. I would put some values of correlation, that is some median values

Methods. It is not clear it patients were enrolled prospectively (consecutive pts) or not

Methods/results/discussion: Inclusion of 24/108 patients (although it is ok sense for a high quality study) probably may limit clinical applicability of the model to a wide range of patients

Methods. The methods part is too long. Maybe some part should be put in appendix (cct evaluation)

Methods. Do authors evalauted intra observer agreement?

Methods. Results of test for normality should be added

Reviewer #2: Nicely written manuscript and interesting findings and analysis. However, concluding that the correlation between impaired cerebrovascular reactivity and retinal vascular density as being caused by similar atherosclerotic changes should be interpreted with caution as both, ocular blood flow and cerebral blood flow are dependent on the flow in carotid artery which is impaired in those patients.

Reviewer #3: Excellent planning and execution of a great thought for research.

The idea of combining a neurological investigation modality with that of pure ocular investigation is brilliant.

My only critique is that this kind of data should be presented in a better looking manner and the results should be highlighted better as this paves the way for more future correlation between neurological and ocular modalities.

Reviewer #4: • This was a prospective, cross-sectional study aimed at correlating several parameters characterizing cerebrovascular reactivity with retinal vascular density changes in an older population with atherosclerotic carotid stenosis. The prospective nature allows adequate application of study methods on all subjects, however it is quite striking that a study that began with 108 patients ended up with 24 patients only. At this point, it seems plausible that the study may not be sufficiently powered anymore to statistically test the several parameters used by the authors in this study. Please include a section in the statistics paragraph to address this issue.

• All the parameters used by the authors do not inherently have normative data. In these types of studies, a healthy cohort of patients are usually used to correlate these figures, which the authors did not do in this study. This is particularly important because the authors determined a decrease in vessel density was seen in these atherosclerotic patients, even though a decline in vessel density is usually a normal ageing sign. A comparison to age-matched individuals may help. Please explain the reasoning behind this.

• The authors used retina vascular density as a marker of volumetric blood changes during carotid compression and release. Vascular density is usually used as a marker or ischemic changes rather than volume difference. Adjusted flow index may have been a more sensitive marker of blood flow changes, as seen in several papers in the literature. This is evident by the authors' sentence 301 in p. 20 that found no correlation between the cerebral indices and retinal OCT values, which they, rather mistakenly, label "retinal hypoperfusion. Please comment.

• The authors do not report in detail the sizes of the OCTA macular scans they used, as well as the rationale behind dividing the ONH vasculature into papillary and peripapillary, and avoiding 6x6 mm ONH scans. Please explain.

• The authors mention they used a "small vessel identification" software that identified those as less than 35 microns. Please explain the significance of this number and the reasoning behind adding this parameter.

• The "cerebrovascular reactivity" and "calculated cerebral arterial resistance" paragraphs in page 15 read more like figure legends and don't necessarily add much to the results section.

• The manuscript contains several linguistic and grammar errors, and may benefit from thorough proof-reading.

6. PLOS authors have the option to publish the peer review history of their article (what does this mean?). If published, this will include your full peer review and any attached files.

Reviewer #1: **Yes: **Fabrizio D'Ascenzo

Reviewer #2: **Yes: **Amr Wassef

Reviewer #3: **Yes: **Hesham Abdelaziz

Reviewer #4: No

---

## [Author Response · Author response to Decision Letter 0]

16 Jun 2023

Responses to Editor and Reviewers

Ayman Elnahry 16 June 2023

Academic Editor 

PLOS ONE

Dear Ayman Elnahry,

We would like to thank you for your editorial evaluation of the manuscript and thank the Reviewers for their critical comments and suggestions, which we believe helped us to improve our manuscript. Please, find our responses to the Editorial letter below.

1. Editor comment 1#: Please ensure that your manuscript meets PLOS ONE's style requirements, including those for file naming.

- We updated the formatting style of the manuscript as it is presented in the template forms, including the instruction of file naming.

2. Editor comment 2#: We note that you have provided funding information that is currently declared in your Funding Statement. However, funding information should not appear in the Acknowledgments section or other areas of your manuscript.

- In the revised manuscript we deleted the acknowledgments and funding sections, and we have updated the funding statement in the cover letter:

“PS: The study was supported by the National Office for Research, Development and Innovation (Project no. NKFIK129277 ("Evaluation of cerebrovascular events in patients with occlusive carotid artery disorders based on morphological and hemodynamic features") has been implemented with the support provided by the Ministry of Innovation and Technology of Hungary from the National Research, Development, and Innovation Fund. 

AK: NKFI-1 K OTKA 132596 K-19, TKP2021-EGA-37 of MIT of Hungary-NRDI TKP2021-EGA funding and HAS/MTA Post-Covid 2021-34. 

DB: Ministry of Innovation and Technology of Hungary from the National Research, Development and Innovation Fund (Grant No: TKP2021-EGA/TKP2021-NVA/TKP2021-NKTA).

3. Editor comment 3#: We require you to either (1) present written permission from the copyright holder to publish these figures specifically under the CC BY 4.0 license, or (2) remove the figures from your submission:

- Figure 2 is an original OCTA photo taken in our laboratory. We have uploaded the Content Permission Form. Please, let us know if you have any questions. 

Please find our responses to the Reviewers' comments below:

Reviewer1#

We would like to thank you for your careful reading and useful comments. We believe they helped to improve the quality of the manuscript. Thank you for your consideration.

Reviewer #1 Point 1#: title: authors should add if the present is a single center study or not 

Author’s response: Title has been updated: “Impaired cerebrovascular reactivity correlates with reduced retinal vessel density in patients with carotid artery stenosis: cross-sectional, single center study”

Reviewer #1 Point 2#: Abstract. I would put some values of correlation, that is some median values 

Author’s response: In the Abstract the results section has been updated: “Twenty-four ICA stenotic patients were evaluated. Both CVR and retinal VD were characterized. There was a significant, negative correlation between CAR-THRR (median=-0.40) and VDPPsmall vessel type (median=52%), as well as between VDPPall vessel types (median=58%), and also similar correlation between CAR-THRR and VDP-WIsmall (median=49.5%) and between VDP-WIall (median=55%).“

Reviewer #1 Point 3#: Methods. It is not clear it patients were enrolled prospectively (consecutive pts) or not. 

Author’s response: Patients were enrolled prospectively and consecutively. In the Methods we added this information: Page 6, line 83: “Patients with significant ICA stenosis were evaluated at the Department of Vascular and Endovascular Surgery of Semmelweis University after providing them with detailed oral information and gaining their written consent, were prospectively and consecutively enrolled between 01.01.2019 and 30.09.2021, and data processing began after patient enrollment.”

Reviewer #1 Point 4#: Methods/results/discussion: Inclusion of 24/108 patients (although it is ok sense for a high quality study) probably may limit clinical applicability of the model to a wide range of patients. 

Author’s response: Yes, indeed, the Reviewer raised an important issue. In response to it we have insert the following text in the revised manuscript: 

Page 24, line 406: “It is of note that both TCD and OCTA examinations have technical limitations. (1-6) However, it is also considered as the strength of our study, that we applied two very specific non-invasive techniques in the same patients. These, however, came with some limitations, but on the other hand - we believe - it makes our study unique. However, loss of exclusion of patients was mainly due to strict inclusion criteria of the study protocol. Also, 17% of patients decline the examinations, because it would require extended hospital stay. 

In addition, the study period coincided with the COVID-19 pandemic affecting Hungary, as well and coinciding with lockdown periods. These factors further reduced the number of potentially eligible patients in the present study.”

Page 25, Line 424: “Despite the mentioned limitations, based on the proven significant correlation, the two test methods complement and may even replace each other, so when one method cannot be used for technical reasons, the other can provide clinical data of similar value.“

Reviewer #1 Point 5#: Methods. The methods part is too long. Maybe some part should be put in appendix (cct evaluation) 

Author’s response: We agree with the Reviewer and thus some parts of the TCD and OCTA descriptions and protocols have been moved to the supporting information:

S1 Appendix. Transcranial Doppler (TCD) study protocol. The text summarizes the technical description of TCD measures.

S2 Appendix. Data processing. The text summarizes the software data processing.

S3 Appendix. Common carotid artery compression test. The text summarizes the description of the standardized common carotid artery compression test.

S4 Appendix. OCTA small vessel criteria. The text summarizes the description of the software-based small vessel definition.

Reviewer #1 Point 6#: Methods. Do authors evaluated intra observer agreement? 

Author’s response: Intra- and inter-observer agreement was performed regarding the TCD measures. The intra-observer agreement of OCTA measurements was not evaluated in this study. However, several previous studies assessed the repeatability and reliability of OCTA measurements and found highly accurate. (3, 4, 7) Regarding the OCTA images the software evaluates the image quality (scan quality - SQ) of the OCTA recordings on a 10-point scale, and only test results with an SQ value exceeding 5 points were included in the study.

Reviewer #1 Point 7#: Methods. Results of test for normality should be added.

Author’s response: In Methods we added: Page 12, line 201 “The normality test was performed with the Kolmogorov-Smirnov test, based on which the data showed a non-normal distribution.”

Reviewer2#

We would like to thank you for your careful reading our manuscript and for your useful comments. We believe they helped to improve the quality of the manuscript. Thank you for your considerations.

Reviewer #2 Point 1#: Nicely written manuscript and interesting findings and analysis. However, concluding that the correlation between impaired cerebrovascular reactivity and retinal vascular density as being caused by similar atherosclerotic changes should be interpreted with caution as both, ocular blood flow and cerebral blood flow are dependent on the flow in carotid artery which is impaired in those patients.

Author’s response: Thank you for pointing out this issue. Yes, indeed, one of the objectives of our study was to assess the carotid artery region, such as the middle cerebral artery and the retinal vasculature, and whether there is a relationship between their blood flow variables. Similar atherosclerotic changes of the two system could be one factor explaining the correlation beside of the hypoperfusion caused by carotid stenosis. We agree, both decreased retinal vascular density and impaired cerebrovascular reactivity are the consequences of the altered blood flow due to atherosclerotic changes in the carotid artery. To illustrate this point, we added a new figure in the manuscript showing the common origin of cerebral and retinal vessels. (Figure will be uploaded in the required resolution to Manuscript Manager)

Figure 1. Schematic depiction of cerebral and retinal circulation affected of atherosclerotic stenosis of the internal carotid artery. ICA= internal carotid artery; MCA= middle cerebral artery (created with www.biorender.com)

Reviewer3#

We would like to thank you for your careful reading of our manuscript and for your useful comments. We believe they helped to improve the quality of the manuscript. Thank you for your considerations.

Reviewer #3 Point 1#: My only critique is that this kind of data should be presented in a better-looking manner and the results should be highlighted better as this paves the way for more future correlation between neurological and ocular modalities. 

Author’s response: Thank You for the helpful suggestions. To illustrate this point, we added a new figure in the manuscript showing the common origin of cerebral and retinal vessels. (Figure will be uploaded in the required resolution to Manuscript Manager)

Figure 1. Schematic depiction of cerebral and retinal circulation affected of atherosclerotic stenosis of the internal carotid artery. ICA= internal carotid artery; MCA= middle cerebral artery (created with www.biorender.com)

Thank you very much, we agree with the Reviewer comment. Thus, to better highlight the results Table 1, Table 2 and Table 3 have been formally re-edited according to PLOS One table formatting rules: cell shading, arrangement of cell content and heavy gridlines.

Reviewer4#

We would like to thank you for your careful reading and useful comments. We believe they helped to improve the quality of the manuscript. Thank you for your consideration.

Reviewer #4 Point 1#: The prospective nature allows adequate application of study methods on all subjects, however it is quite striking that a study that began with 108 patients ended up with 24 patients only. At this point, it seems plausible that the study may not be sufficiently powered anymore to statistically test the several parameters used by the authors in this study. Please include a section in the statistics paragraph to address this issue. 

Author’s response: Yes, indeed, the Reviewer raised an important issue. In response to it we have insert the following text in the revised manuscript: Page 24, line 406: “It is of note that both TCD and OCTA examinations have technical limitations. (1-6) However, it is also considered as the strength of our study, that we applied two very specific non-invasive techniques in the same patients. These, however, came with some limitations, but on the other hand - we believe - it makes our study unique. However, loss of exclusion of patients was mainly due to strict inclusion criteria of the study protocol. Also, 17% of patients decline the examinations, because it would require extended hospital stay. 

In addition, the study period coincided with the COVID-19 pandemic affecting Hungary, as well and coinciding with lockdown periods. These factors further reduced the number of potentially eligible patients in the present study.”

As requested by the reviewer the following paragraph was added to the description of statistical methods at page 12 line 203: “The minimum sample size was determined by statistical power calculation (power 0.80; p = 0.05) using data from previous studies at our institution and the method proposed by Hulley et al. for correlation analyzes (8) The minimum number of eyes to be included in this study was calculated to be 22 eyes to provide sufficient power for bivariate correlation analyses.”

However, as the power analysis is only just one aspects of the interpretation of the test results and considering the complexity of the entire study design, we consider our results to be sufficient despite of this limitation, nevertheless, emphasizing that further studies are necessary to generalize the findings of our study. 

Reviewer #4 Point 2#: All the parameters used by the authors do not inherently have normative data. In these types of studies, a healthy cohort of patients are usually used to correlate these figures, which the authors did not do in this study. This is particularly important because the authors determined a decrease in vessel density was seen in these atherosclerotic patients, even though a decline in vessel density is usually a normal ageing sign. A comparison to age-matched individuals may help. Please explain the reasoning behind this. 

Author’s response: Indeed, the reviewer raised a very logical questions regarding the age and atherosclerosis. Thank you for this useful comment, and to be responsive, the following new data were inserted into the revised manuscript.

Page 23, line 385: “Regarding the OCTA studies previous studies on the same patient group described the decrease of retinal vessel density in this patient group. (9) “

Page 16, line 250: “To contrast these results to healthy vascular system in the present study five age-matched control participants (mean age = 63.2 ± 7.1) with an absence of atherosclerotic carotid artery disease and other major vascular risk factors were recruited. Statistical analysis showed a significant difference in CAR-THRR between the control and study group (Mann-Whitney U test, p=0.045). “

Nevertheless, in the future studies more participants in both the control group and the study group will be investigated, providing a wider applicability of our results. 

Reviewer #4 Point 3#: The authors used retina vascular density as a marker of volumetric blood changes during carotid compression and release. Vascular density is usually used as a marker or ischemic changes rather than volume difference. Adjusted flow index may have been a more sensitive marker of blood flow changes, as seen in several papers in the literature. This is evident by the authors' sentence 301 in p. 20 that found no correlation between the cerebral indices and retinal OCT values, which they, rather mistakenly, label "retinal hypoperfusion. Please comment. 

Author’s response: Thank you for this comment. Hypoperfusion corresponds to MCA. The retinal vessel density was statical measure, that was correlated with a functional TCD test (CCC test). The OCT device used in the research is not suitable for calculating the adjusted flow index, however, in the future it would really be worthwhile to examine the changes of this parameter in this patient group as well.

The reviewer point is right, thus in order to avoid the potential interference, the CCC test and OCTA evaluation were performed separately. To clarify this, we deleted the term “retinal hypoperfusion” of the revised manuscript. 

Reviewer #4 Point 4#: The authors do not report in detail the sizes of the OCTA macular scans they used, as well as the rationale behind dividing the ONH vasculature into papillary and peripapillary, and avoiding 6x6 mm ONH scans. Please explain.

Author’s response: 3x3 mm scans were obtained from the central macular area, which is the smallest scan size among the protocols. We also chose the smallest possible scan size (4.5x4.5 mm) for the peripapillary area, as smaller scans proved to have a higher resolution and are therefore more suitable for quantitative analysis. Previous studies found a decrease of the peripapillary vessel density values in patients with cerebral small vessel disease (10, 11), therefore, in addition to analyzing the vessel density of the whole image we also examined the peripapillary vessel density separately. 

Reviewer #4 Point 5#: The authors mention they used a "small vessel identification" software that identified those as less than 35 microns. Please explain the significance of this number and the reasoning behind adding this parameter.

Author’s response: The built in software of the OCT device automatically differentiates the small vessels based on the user manual, vessels smaller than 35 microns are classified in this category. This information is added as supporting information in the revised manuscript.

Reviewer #4 Point 6#: The "cerebrovascular reactivity" and "calculated cerebral arterial resistance" paragraphs in page 15 read more like figure legends and don't necessarily add much to the results section.

Author’s response: Thank you for your recommendation, accordingly we updated the manuscript and the two subheadings have been merged. 

In text at page 15 line 223: “Cerebrovascular reactivity and calculated cerebral arterial resistance"

Reviewer #4 Point 7#: The manuscript contains several linguistic and grammar errors, and may benefit from thorough proof-reading. 

Author’s response: Thank you for your suggestion, in order to improve the comprehensibility of the text, we had fully revised our manuscript, which was then further corrected by an English proof-reader before re-submission. 

References

1. Purkayastha S, Sorond F. Transcranial Doppler ultrasound: technique and application. Semin Neurol. 2012;32(4):411-20.

2. Enders C, Lang GE, Dreyhaupt J, Loidl M, Lang GK, Werner JU. Quantity and quality of image artifacts in optical coherence tomography angiography. PLoS One. 2019;14(1):e0210505.

3. Lee MW, Kim KM, Lim HB, Jo YJ, Kim JY. Repeatability of vessel density measurements using optical coherence tomography angiography in retinal diseases. Br J Ophthalmol. 2018.

4. Lei J, Durbin MK, Shi Y, Uji A, Balasubramanian S, Baghdasaryan E, et al. Repeatability and Reproducibility of Superficial Macular Retinal Vessel Density Measurements Using Optical Coherence Tomography Angiography En Face Images. JAMA Ophthalmol. 2017;135(10):1092-8.

5. Sharma VK, Wong KS, Alexandrov AV. Transcranial Doppler. Front Neurol Neurosci. 2016;40:124-40.

6. Willie CK, Colino FL, Bailey DM, Tzeng YC, Binsted G, Jones LW, et al. Utility of transcranial Doppler ultrasound for the integrative assessment of cerebrovascular function. J Neurosci Methods. 2011;196(2):221-37.

7. Rabiolo A, Gelormini F, Sacconi R, Cicinelli MV, Triolo G, Bettin P, et al. Comparison of methods to quantify macular and peripapillary vessel density in optical coherence tomography angiography. PLoS One. 2018;13(10):e0205773.

8. Hulley SB CS, Browner WS, Grady D, Newman TB. Designing clinical research: an epidemiologic approach. 4th ed. ed. Gaertner R, editor. Philadelphia, PA 19103 USA: Lippincott Williams & Wilkins, a Wolters Kluwer; 2013. 79 p.

9. István L, Czakó C, Benyó F, Élő Á, Mihály Z, Sótonyi P, et al. The effect of systemic factors on retinal blood flow in patients with carotid stenosis: an optical coherence tomography angiography study. Geroscience. 2022;44(1):389-401.

10. Lee JY, Kim JP, Jang H, Kim J, Kang SH, Kim JS, et al. Optical coherence tomography angiography as a potential screening tool for cerebral small vessel diseases. Alzheimers Res Ther. 2020;12(1):73.

11. Wang X, Wei Q, Wu X, Cao S, Chen C, Zhang J, et al. The vessel density of the superficial retinal capillary plexus as a new biomarker in cerebral small vessel disease: an optical coherence tomography angiography study. Neurol Sci. 2021;42(9):3615-24.

We thank you for your comments and questions you raised. We believe they helped to improve the quality of the manuscript.

Sincerely yours, 

Rita Magyar-Stang

Semmelweis University,

Department of Neurology

Balassa János Street 6.

1083, Budapest, Hungary

Tel: +36-20/666-3274

E-mail: stang.rita@semmelweis.hu

---

## [Decision Letter · Decision Letter 1]

31 Aug 2023

Impaired cerebrovascular reactivity correlates with reduced retinal vessel density in patients with carotid artery stenosis: cross-sectional, single center study

PONE-D-23-12728R1

Dear Dr. Magyar-Stang,

We’re pleased to inform you that your manuscript has been judged scientifically suitable for publication and will be formally accepted for publication once it meets all outstanding technical requirements.

Kind regards,

Ayman Elnahry

Academic Editor

PLOS ONE

Additional Editor Comments (optional):

Thank you for responding to the reviewers’ comments. Please correct the minor spelling error that the reviewers pointed out in the proof phase of publication.

Reviewers' comments:

Reviewer's Responses to Questions

**Comments to the Author**

1. If the authors have adequately addressed your comments raised in a previous round of review and you feel that this manuscript is now acceptable for publication, you may indicate that here to bypass the “Comments to the Author” section, enter your conflict of interest statement in the “Confidential to Editor” section, and submit your "Accept" recommendation.

Reviewer #1: All comments have been addressed

Reviewer #2: All comments have been addressed

Reviewer #3: (No Response)

Reviewer #4: All comments have been addressed

2. Is the manuscript technically sound, and do the data support the conclusions?

Reviewer #1: (No Response)

Reviewer #2: Partly

Reviewer #3: Yes

Reviewer #4: Yes

3. Has the statistical analysis been performed appropriately and rigorously? 

Reviewer #1: (No Response)

Reviewer #2: I Don't Know

Reviewer #3: Yes

Reviewer #4: Yes

4. Have the authors made all data underlying the findings in their manuscript fully available?

Reviewer #1: (No Response)

Reviewer #2: No

Reviewer #3: Yes

Reviewer #4: Yes

5. Is the manuscript presented in an intelligible fashion and written in standard English?

Reviewer #1: (No Response)

Reviewer #2: Yes

Reviewer #3: Yes

Reviewer #4: Yes

6. Review Comments to the Author

Reviewer #1: (No Response)

Reviewer #2: (No Response)

Reviewer #3: 266 There is a spelling mistake of OTCA instead of OCTA

Reviewer #4: I think the authors responded well to the comments. I'd still like to point out the typo in page 17, line 266 to be corrected to "OCTA". Otherwise, I don't have more to add.

7. PLOS authors have the option to publish the peer review history of their article (what does this mean?). If published, this will include your full peer review and any attached files.

Reviewer #1: **Yes: **Fabrizio D'Ascenzo

Reviewer #2: **Yes: **Amr Wassef

Reviewer #3: No

Reviewer #4: No

---

## [Editor Report · Acceptance letter]

6 Sep 2023

PONE-D-23-12728R1 

Impaired cerebrovascular reactivity correlates with reduced retinal vessel density in patients with carotid artery stenosis: cross-sectional, single center study 

Dear Dr. Magyar-Stang:

I'm pleased to inform you that your manuscript has been deemed suitable for publication in PLOS ONE. Congratulations! Your manuscript is now with our production department. 

Kind regards, 

on behalf of

Dr. Ayman Elnahry 

Academic Editor

PLOS ONE